# Roles of Oxygen Vacancies of CeO_2_ and Mn-Doped CeO_2_ with the Same Morphology in Benzene Catalytic Oxidation

**DOI:** 10.3390/molecules26216363

**Published:** 2021-10-21

**Authors:** Min Yang, Genli Shen, Qi Wang, Ke Deng, Mi Liu, Yunfa Chen, Yan Gong, Zhen Wang

**Affiliations:** 1School of Chemistry and Biological Engineering, University of Science and Technology, Beijing 100083, China; yangmin@ustb.edu.cn; 2CAS Key Laboratory of Standardization and Measurement for Nanotechnology, CAS Center for Excellence in Nanoscience, National Center for Nanoscience and Technology, Beijing 100190, China; shengl@nanoctr.cn (G.S.); wangq@nanoctr.cn (Q.W.); kdeng@nanoctr.cn (K.D.); liumi@nanoctr.cn (M.L.); 3State Key Laboratory of Multiphase Complex Systems, Institute of Process Engineering, Chinese Academy of Sciences, Beijing 100190, China; yfchen@home.ipe.ac.cn

**Keywords:** Mn-doped CeO_2_, pure CeO_2_, morphology, oxygen vacancy, benzene oxidation

## Abstract

Mn-doped CeO_2_ and CeO_2_ with the same morphology (nanofiber and nanocube) have been synthesized through hydrothermal method. When applied to benzene oxidation, the catalytic performance of Mn-doped CeO_2_ is better than that of CeO_2_, due to the difference of the concentration of O vacancy. Compared to CeO_2_ with the same morphology, more oxygen vacancies were generated on the surface of Mn-doped CeO_2_, due to the replacement of Ce ion with Mn ion. The lattice replacement has been analyzed through XRD, Raman, electron energy loss spectroscopy and electron paramagnetic resonance technology. The formation energies of oxygen vacancy on the different exposed crystal planes such as (110) and (100) for Mn-doped CeO_2_ were calculated by the density functional theory (DFT). The results show that the oxygen vacancy is easier to be formed on the (110) plane. Other factors influencing catalytic behavior have also been investigated, indicating that the surface oxygen vacancy plays a crucial role in catalytic reaction.

## 1. Introduction

Volatile organic compounds (VOCs) are a type of low-boiling point organic matter which contains aromatic hydrocarbons, straight-chain alkane and cycloalkanes, etc. [1]. VOCs can be produced in industrial process and daily life and bring harm to human health and environment. Hence, it is particularly urgent to limit VOC emissions [2]. There are many methods to remove VOCs, in which catalytic oxidation is considered to be the most effective method due to its destructive efficiency, lower operating temperature and less by-products [3,4]. At present, two types of common catalysts have been widely researched: one is precious metal type and the other is transition metal oxide type. The application of former is limited due to the cost, sintering risk and facile poisoning. In contrast, the transition metal oxide is attracting more attention owing to the high-stability, low cost and high oxidation activity at low temperatures [5,6,7]. In recently years, transition metal based oxides catalysts exhibited excellent catalytic behaviors towards the removal of VOCs, such as Mn-based [8,9,10] and Co-based [11,12] metal oxide catalysts.

Additionally, rare-earth oxides have also attracted wide attention over the VOCs catalytic oxidation. CeO_2_, as typical rare earth oxide has been investigated in the field of heterogeneous catalysis and exhibits enhanced performances for VOC oxidation due to its outstanding redox property and oxygen storage capacity (OSC). It is worthy to note that the microstructure of CeO_2_ is closely related with the catalytic activity, especially crystal plane defects. Trovarelli and Llorca [13] reported different planes of CeO_2_ possessed different content of oxygen vacancy. The activity of CeO_2_ for VOC oxidation is sensitive to the proportion of surface oxygen defects in the catalyst [14]. Once the surface active oxygen vacancies are covered by intermediates or carbon deposition, the catalytic behavior of CeO_2_ will be poisoned [15,16]. In order to increase the content of active sites, transition metals are often used to be doped or composited with CeO_2_ to produce more oxygen vacancies. Among the transition metals doped catalysts, Mn-based catalysts are widely used in the field of environmental catalysis [17,18]. It is reported that Mn-dopedCeO_2_has been applied in the abatement of contaminants, such as the catalytic reduction of NO and diesel soot combustion, which exhibit excellent catalytic activity [19,20,21,22,23].

In this article, we have investigated the catalytic behavior of pure CeO_2_ and Mn-doped CeO_2_ with the same morphology through the complete catalytic oxidation of benzene. Their catalytic activities differences and crystal structures are analyzed in detail. Meanwhile, density function theory (DFT) was adopted to simulate crystal plane structure and calculate the formation energy of oxygen vacancies on the certain exposed plane. Although the effect of oxygen vacancy in pure CeO_2_ on the catalytic property has been recognized, however the influence of dopants in oxygen vacancy for doped CeO_2_ need to be described further. In addition, the difference of oxygen vacancy on the different exposed crystal plane has less been studied. Therefore, the research aims to explore the effect of crystal plane structure and dopant (Mn ion) dispersion on the formation of oxygen vacancy and analyze the roles of oxygen vacancy in the combustion of benzene further.

## 2. Materials and Methods

### 2.1. The Preparation of CeO_2_-MnO_x_ Composite Oxides

The chemicals used in this work, including Ce(NO_3_)_3_·6H_2_O (99%), Mn(NO_3_)_2_ solution (50%), NaOH (98%), and ethanol, were purchased from Beijing Chemicals Company (Beijing, China). Mn-doped CeO_2_ with different morphologies (nanofiber: NF and nanocube: NC) were synthesized by a hydrothermal process. Firstly, Ce(NO_3_)_3_·6H_2_O and Mn(NO_3_)_2_ in given amount (Ce/Mn mole ratio = 9/1) were dissolved in a 10 mL H_2_O and mixed with a 6 M NaOH solution. The solution was stirred for 0.5 h at room temperature then transferred to an autoclave (100 mL) and gradually heated to a certain temperature, e.g., 120 °C for NF while 160 °C for NC, respectively. The reaction at the targeting temperature was kept for 24 h. After the reaction, the precipitates in autoclaves were collected by centrifugation, washed with distilled water and ethanol several times. The obtained materials, labeled as Ce-Mn-NF and Ce-Mn-NC respectively, were dried at 80 °C overnight and calcined at 550 °C for 4 h. Pure CeO_2_with fiber and cube morphology were also prepared using the similar process as reference, labeled as CeO_2_-NF and CeO_2_-NC respectively.

### 2.2. Characterization

The crystal phases of the catalysts were characterized by X-ray diffraction (XRD) using PhilipsX’pert PRO analyzer (Philips, Amsterdam, The Netherlands) equipped with a Cu Kα radiation source (λ = 0.154187 nm) at a scanning rate of 0.03°/s (2θ from 10° to 90°).

Scanning electron microscopy (SEM JC-Zeiss Merlin) and Transmission electron microscopy (TEM Tecnai G2 F20 U-TWIN) was used to observe the morphology and structure of samples. Aberration-corrected HAADF-STEM images, Mn L_2,3_-edge and Ce M_4,5_-edge electron energy loss spectroscopy (EELS) were performed from FEI Titan electron microscope equipped with a Gatan Enfnium camera system.

The surface composition and chemical states were determined by X-ray photoelectron spectroscopy (XPS ESCALAB250Xi). The binding energy (BE) was calibrated using the C1s line at 284.8 eV.

Raman spectra were measured with a Raman spectrometer (Renishaw inVia plus). The excitation source was an Ar^+^ ion laser (λ = 514.23 nm) and the laser power was 20 mW.

Reactive oxygen species were identified by the electron paramagnetic resonance technology (EPR) at room temperature on a Bruker EMX spectrometer (Bruker Corp., Billerica, MA, USA) at a frequency of 9.8 GHz and a magnetic field of 100 kHz.

Hydrogen temperature-programmed reduction (H_2_-TPR) was performed on AutoChem 2920 chemisorption analyzer. Firstly, the sample (40–60 mesh) was heated to 150 °C and maintained at this temperature for 1h in 5% O_2_ and 95% He mixture (30 mL/min), and then cooled to 50 °C under He flow. The samples were then heated again to 900 °C in 10% H_2_ and 90% Ar mixture (50 mL/min). The heating rate is 10°C/min. The thermal conductivity detector signal was detected.

### 2.3. Catalytic Activity Tests

Activity tests for catalytic oxidation of benzene over CeO_2_-NF, CeO_2_-NC, Ce-Mn-NF and Ce-Mn-NC catalysts were performed in a continuous-flow fixed-bed reactor, containing 100 mg catalyst samples (40–60 mesh), respectively. The total reactant flow rate was 100 mL·min^−1^, containing (100 ppm benzene + 20 vol% O_2_ + N_2_ (balance)). The weight hourly space velocity (WHSV) was typically 60,000 mL·g^−1^·h^−1^. The concentrations of Benzene, CO_2_, and CO were analyzed on-line using a gas GC/MS (Hewlett-Packard 6890N and Hewlett-Packard 5973N). 5.0 vol% water vapor was introduced to the system by a water saturator at 34 °C. Catalytic activities were characterized by two parameters (T_50_ and T_90_). For a catalyst, T_50_ and T_90_ represent the temperature of 50 and 90% benzene conversion, respectively. The benzene conversion and CO_2_ selectivity are calculated as following Equations (1) and (2). [Benzene]*_in_* and [Benzene]*_out_* are the inlet and outlet Benzene concentrations during catalytic reaction):(1)$$\mathrm{Benzene}\mathrm{conversion}(\%)=\frac{{\left[\mathrm{Benzene}\right]}_{in}-{\left[\mathrm{Benzene}\right]}_{out}}{{\left[\mathrm{Benzene}\right]}_{in}}\times 100\%$$
(2)$${\mathrm{CO}}_{2}\mathrm{selectivity}(\%)=\frac{\left[{\mathrm{CO}}_{2}\right]}{\left(\left[\mathrm{CO}\right]+\left[{\mathrm{CO}}_{2}\right]\right)}\times 100\%$$

### 2.4. Details of DFT + U Calculation

First-principles calculations based on density functional theory (DFT) were carried out with the Vienna Ab-initio simulation package (VASP) and PBE functional. The interaction between core electrons and valence electrons was expressed by the projector-augmented wave (PAW) method. The cutoff energy of the plane-wave basis set was set to 550 eV. To guarantee the accuracy, the convergence criteria of energy and force were set to 10^−5^ eV and 0.05 eV·Å^−1^, respectively. DFT + U with U = 5eV was applied to treat Ce 4f orbital. A model was built based on a 2 × 2 × 2 CeO_2_ supercell. The Brillouin zone of (100) and (110) models were sampled by a 3 × 3 × 1 k-point set to acquire similar sampling densities. Before calculation, no more than half of the atomic layers from the bottom were fixed.

## 3. Results and Discussion

### 3.1. Characterization of Catalysts

The crystal structures of CeO_2_-NF, CeO_2_-NC, Ce-Mn-NF and Ce-Mn-NC was examined by XRD. As seen in Figure 1a, the diffraction peaks at 2θ = 28.5°, 33.0°, 47.4°, 56.4°, 59.2°, 69.5°, 76.6°, 79.1° and 88.6° clearly demonstrate the presence of cubic fluorite structure of CeO_2_ (PDF# 81-0792). For Mn-doped CeO_2_ (Ce-Mn-NF and Ce-Mn-NC), the patterns do not show any diffraction of manganese oxides, and only broad reflections are observed, which should be attributed to the formation of Ce-O-Mn solid solution. It is worthy to note that the characteristic diffraction peak of CeO_2_ (2θ = 28.5°) in the Mn-doped CeO_2_ moves to little bit higher value due to the decrease of the lattice parameter caused by the dopant Mn ions with smaller size, compared with the pure CeO_2_ (Figure 1b).

The lattice parameters of all the samples have been calculated according to the XRD data and listed in Table 1. The result shows that the lattice parameters of the Mn-doped CeO_2_ are lower than those of pure CeO_2_ with the same morphology. The ionic radius of Mn^n+^ {Mn^2+^ (0.067nm), Mn^3+^ (0.066 nm) and Mn^4+^ (0.053 nm)} are all smaller than that of the Ce^4+^ (0.1098 nm), therefore, the incorporation of Mn^n+^ into the CeO_2_ lattice to form Ce-O-Mn solid solution could result in lattice contraction due to the doping of Mn^n+^ and the consequent formation of the more oxygen vacancies.

The morphology and structure of samples were characterized by SEM and TEM, respectively. Figure 2a shows the SEM image of CeO_2_-NF. It can be seen that CeO_2_-NF with a fiber morphology has an average diameter of about 10 nm and length of about 150–300 nm. In the HRTEM of CeO_2_-NF (Figure 2b), the exposed crystal planes (110) and (100) can be determined obviously according to the inter-planar distance 0.19 nm and 0.27 nm, respectively. Figure 2c shows the SEM image of CeO_2_-NC. The size of CeO_2_-NC possessing cubic morphology is about 12–25 nm. Through the HRTEM (Figure 2d), it can be acquired that the exposed crystal plane is only (100). (111) plane also exists inside the lattice. For Ce-Mn-NF and Ce-Mn-NC, the morphology has no change compared with pure CeO_2_ with the same morphology except the change of sample size (Figure 2e–h). In order to analyze the element distribution of Mn-doped CeO_2_, the HAADF mapping image of Ce-Mn-NF and Ce-Mn-NC were examined, respectively (Figure 2i,j). The result exhibits that both the Ce and Mn elements are homogeneously dispersed, which favors the oxygen vacancy formation.

Raman scattering was used to identify the solid solution phase, which can reflect indirectly the property of oxygen vacancy. Figure 3 shows visible Raman spectra of all the samples. In the pattern, a strong peak at ca. 460 cm^−1^ and a weak one at ca. 610 cm^−1^ are discerned, corresponding to the fluorite F_2g_ mode and a defect-induced mode (D mode), respectively [24,25]. The peaks at 275 and 1163 cm^−1^ are assigned to second-order transverse and longitudinal vibration modes of cubic CeO_2_ fluoride phase [26]. These bands at ca. 610 cm^−1^are usually assigned to the presence of extrinsic oxygen vacancies created as charge compensating defects during solid solution formation [27]. For Ce-Mn-NF and Ce-Mn-NC, a band at ca. 610 cm^−1^ corresponds to the second-order Raman mode attributes of O^2−^ vacancies formed by a low valence dopant (Mn^n+^). For pure CeO_2_, Raman band is also detected at ca. 610 cm^−1^, which is also assigned to the presence of oxygen vacancies resulted from Ce^4+^ transforming to Ce^3+^. The ratio of integrated peak area of oxygen vacancy (~610 cm^−1^) to that of main peak (460 cm^−1^), defined as A_610_/A_460_, is used here to characterize the relative amount of oxygen vacancy among these samples. The A_610_/A_460_ ratio was calculated and ranked in the order of Ce-Mn-NF> Ce-Mn-NC> CeO_2_-NF> CeO_2_-NC (Table 2), suggesting that Ce-Mn-NF exhibits the higher concentration of oxygen vacancies.

The oxidation state of catalyst surface species was examined by XPS analysis. Figure 4a presents the XPS spectra of the Ce3d core levels for all samples. The peaks V_0_/ V_0′_, V_1_/V_1′_and V_2_/V_2′_ refer to three pairs of spin-orbit doublets, which can be attributed to surface Ce^4+^ [28], while U_0_/U_0′_ andU_1_/U_1′_ can be ascribed to Ce^3+^ [29]. The relative amount of Ce^3+^ for all the samples can be calculated according to the Equation (3) and listed in Table 2. The result shows that the relative atomic ratio of Ce^3+^/(Ce^3+^ + Ce^4+^) in the Ce-Mn-NF is higher than that of CeO_2_-NF. The amount of Ce^3+^ in the Ce-Mn-NC is also higher than that of CeO_2_-NC. The data indicate that the introduction of Mn ion can increase the relative amount of Ce^3+^ in the Mn-doped CeO_2_, which might arise from the charge rebalancing of oxygen vacancies and dopant Mn^n+^ [30].
(3)XCe3+=ACe3+SCe∑A(Ce3++Ce4+)SCe×100%
where $X_{Ce^{3+}}$ is the percentage content of Ce^3+^, A is the integrate area of characteristic peak in the XPS pattern, S is sensitivity factors (S = 7.399).

Figure 4b depicts the O 1s XPS spectra of the CeO_2_-NF, CeO_2_-NC, Ce-Mn-NF and Ce-Mn-NC samples. All catalysts exhibited dominant component of lattice oxygen (O_α_) species, together with a shoulder of surface adsorbed oxygen (O_β_) species on the surface vacancies. The binding energy of 529.0–530.0 eV corresponded to O_α_ and the binding energy of 531.0–532.0 eV was ascribed to O_β_. The integral ratio of (O_β_/O_α_+ O_β_) was applied to estimate the concentration of adsorbed oxygen species (Table 2). Ce-Mn-NF and Ce-Mn-NC samples possess more surface absorbed oxygen species compared with CeO_2_-NF and CeO_2_-NC, which indicate that Mn-doped CeO_2_ contain more surface oxygen vacancy. Hence, it is inferred that the surface oxygen vacancy adsorbs and activates oxygen molecules to produce adsorbed oxygen species. This mechanism promotes the redox property of catalyst. A similar phenomenon was reported in the literature [31]. The O 1s XPS spectra results were in accordance with the XRD and Raman data.

Figure 4c presents the Mn 2p patterns of Ce-Mn-NF and Ce-Mn-NC. The binding energies of the Mn 2p_3/2_ component appear at 641.8 eV and those for Mn 2p_1/2_ appear at 653.5 eV. The spin orbit splitting is ΔE = 11.7 eV and the width is 3.62 eV. According to literature data [32], the observed binding energy is tented to correspond to Mn_2_O_3_. It should be noted, though, that the BEs of various Mn ions are very close to each other, rendering impossible the exact identification of oxidation states due to overlap of the energy ranges for various oxidation states of Mn [33]. In the Figure 4c, the Mn 2p_3/2_ XPS spectra were fitted by the presence of Mn^2+^, Mn^3+^ and Mn^4+^. The binding energy of641.9–643 eV, 641.4–641.9 eV and 640.1–641.31 eV was ascribed to the peaks of Mn^4+^, Mn^3+^ and Mn^2+^ species [34], respectively.

The distribution feature of oxygen vacancy can also be reflected through electron energy loss spectroscopy (EELS). The patterns of CeO_2_ and Mn-doped CeO_2_ are shown in Figure 5a, in which Mn L_2__,3_-edge and Ce M_4,5_-edge spectra can be distinguished obviously. Through the Mn L_2,3_-edge spectrum, it can be obtained that Mn ion exhibits mainly trivalent [35], which is consistent with the XPS data of Mn element. According to the literature [36,37,38], it can be acquired that the formation of oxygen vacancies is coupled with the localization delocalization effect of Ce 4f electrons. Therefore, the Ce M_4,5_-edge EELS spectra were used to further investigate the distribution conditions of Ce^3+^ and oxygen vacancies (Figure 5b). The proportions of Ce^3+^ ([Ce^3+^]) were calculated via the M_5_/M_4_ white-line ratios for every sample [39]. The data show that the relative proportions of Ce^3+^ in Ce-Mn-NF (0.876) and Ce-Mn-NC (0.837) are more than those of CeO_2_ with the same morphology (CeO_2_-NF: 0.775, CeO_2_-NC: 0.765) due to Mn^n+^ replacing some Ce^4+^, which indicate that abundant Ce^3+^ exists on the surface of Mn-doped CeO_2_ and more oxygen vacancies are formed. The conclusion has also been identified by the results of Raman and XPS.

EPR is a sensitive to the paramagnetic species with unpaired electrons [40]. Thus, EPR spectra of the pure CeO_2_and Mn-doped CeO_2_ samples were collected to study the oxygen vacancies and the species of reactive oxygen (·O_2_^−^, ^1^O_2_ and ·OH) [41,42]. As presented Figure 6a, pure-CeO_2_ (CeO_2_-NF and CeO_2_-NC) exhibited two types of peaks of Ce^3+^ in terms of the surface (g = 1.965) and bulk (g = 1.947). After Mn doping, Mn-doped CeO_2_ presented a strong EPR signal at g = 2.032, ascribed to the combination of Mn ion with one or two electrons [43,44]. The typical·O_2_^-^ quartet spectrum, ^1^O_2_ triplet spectrum and ·OH spectrum can be detected for all the catalysts (Figure 6b–d). The intensities of peaks corresponding to ·O_2_^−^ and ^1^O_2_ respectively in the Mn-doped CeO_2_ catalyst are far higher than those of CeO_2_. This may be due to more Ce^3+^ to Ce^4+^ transition on the catalyst surface of Mn-doped CeO_2_, which provided many electrons to O_2_ and accelerated the conversion O_2_ to active oxygen species. It is beneficial to accelerate the catalytic oxidation of benzene. It is worthy to note that CeO_2_ only exhibit ·OH characteristic quartet peaks, however Mn-doped CeO_2_ contain other peaks attributed to oxidized DMPO (radical scavenger) besides ·OH peaks, which indicate that Mn-doped CeO_2_ possess higher oxidization ability (Figure 6d). Therefore, it is easier to catalytic oxidize benzene for Mn-doped CeO_2_. The EPR patterns of oxidized DMPO, labeled as DMPO-X can be seen in the Appendix A.

The reducible surface oxygen species of CeO_2_ and Mn-doped CeO_2_were studied using H_2_-TPR as illustrated in Figure 7. TheH_2_-TPR measurement of CeO_2_-NF exhibits two major reduction peaks at 478 °C and 725 °C. The two reduction peaks at 480 °C and 728 °C exist in the pattern of CeO_2_-NC. According to the literature [45,46,47], the former low-temperature reduction (478 °C and 480 °C) is due to the removal of surface oxygen and the later high-temperature reduction (725 °C and 728 °C) is attributed to the release of oxygen species in bulk CeO_2_. For Ce-Mn-NF, the TPR profile exhibits three overlapping peaks at lower temperature and one peak at higher temperature. The lower temperature peaks at 240°Cand 336 °Care assigned to the two-step reduction of Mn_2_O_3_ (Mn_2_O_3_→Mn_3_O_4_; Mn_3_O_4_→MnO) [48], while the reduction peak at 400 °C is corresponded to the reduction of surface oxygen in CeO_2_. The higher temperature peak at 722 °C is attributed to the reduction of oxygen species in bulk CeO_2_. The H_2_-TPR pattern of Ce-Mn-NC is similar with that of Ce-Mn-NC. The lower temperature peaks (200 and 337 °C) correspond to the two-step reduction of Mn_2_O_3_. The peaks at 386 and 743 °C are attributed to the reduction of surface oxygen and bulk oxygen species in CeO_2_, respectively. In addition, the reduction peaks of Mn-doped CeO_2_ are shifted toward lower temperature compared with those of CeO_2_, which indicate the reducibility of Mn-doped CeO_2_ catalyst was increased due to Mn ion. Hence Mn-doped CeO_2_ should possess higher catalytic activities, which is in accordance with the result of EPR.

### 3.2. Catalytic Oxidation Activity towards Benzene

The catalytic activities of Mn-doped CeO_2_ and pure CeO_2_ with the same morphology were evaluated by their oxidation performance towards benzene (Figure 8a). It can be seen that the catalytic activity decreases in the order of Ce-Mn-NF, Ce-Mn-NC, CeO_2_-NF and CeO_2_-NC, which is consistent with that of oxygen vacancy concentration. The oxygen vacancy is closely related with the catalytic activity. T_50_ and T_90_ on Ce-Mn-NF is 278 and 395 °C respectively, which are much lower than those on Ce-Mn-NC (304/450 °C) and pure phase oxide. The catalytic activities of the catalysts were also compared based on the yield of CO_2_. From the Figure 8b, it can be also obtained that the catalytic properties of Mn-doped CeO_2_ are enhanced compared with CeO_2_ with the same morphology. Meanwhile benzene has almost been converted into CO_2_ completely without the detection of any other gas. Water vapor is a common component of industrial waste gas, therefore the impact of water vapor on catalytic activity is requisite. The impact of 5.0 vol% water vapor introduction on catalytic activities of Ce-Mn-NC and Ce-Mn-NF at 400 °C was conducted (Figure 8c,d). As a result, the introduction of 5.0 vol% water vapor leads to benzene conversion dropping from 91% to 77% for Ce-Mn-NC and 83%to 67% for Ce-Mn-NF, respectively. After 10 h of continuous 5.0 vol% water vapor injection, the benzene percent conversion for Ce-Mn-NC and Ce-Mn-NF is completely recovered upon the cutting off of the water vapor feeding. The competitive adsorption between H_2_O and benzene molecules onto the active sites is the main reason causing the decrease of benzene conversion. The result above indicated both of Ce-Mn-NC and Ce-Mn-NF having good water-resistant ability.

### 3.3. DFT Calculations

Through the analysis above, we have recognized that the oxygen vacancy has an important effect on their catalytic activities; however enough theory evidences are still needed. In this article, the density functional theory (DFT) is adopted to calculate the formation energy of oxygen vacancy so that identifying its role in the catalytic reaction further (Figure 9). The defect formation energy (*E_f_*) of oxygen vacancy is defined as Equation (4):(4)$$E_{f}=E_{S\_V_{O}}+\frac{1}{2}E_{O_{2}}-E_{S}$$
where $E_{S\_V_{O}}$ and $E_{S}$ are the total energies of the supercell with and without an oxygen vacancy, and $E_{O_{2}}$ is the total energy of an O_2_ molecule [49].

We compare the oxygen vacancy formation energies (*E_f_*) on the (110) and (100) crystal planes of Mn-doped CeO_2_. The calculated *E_f_* corresponding to (110) plane is -0.48 eV. In contrast, the *E_f_* on the (100) plane increases to 4.46 eV. The result means that the oxygen vacancy is easier to be formed on the (110) plane than (100) plane, which also indicate Mn ion is easier to be doped on the (110) plane. Given the central role of oxygen vacancy in catalyst [50,51], the Ce-Mn-NF [exposed (110) and (100) planes] is therefore expected to show better performance than Ce-Mn-NC [exposed (100) plane] and pure CeO_2_ (CeO_2_-NF and CeO_2_-NC), which is consistent with the results of benzene catalytic degradation. In addition, the distances of Ce-O near the oxygen vacancy in the Mn-doped CeO_2_, whether (110) or (100) planes, have only slight differences compared with those of CeO_2_ which may be caused by the substitution of Mn ion. The distance data of Ce-O bond can be seen in Appendix A.

### 3.4. Factor Influencing the Catalytic Activity

Through the analysis, it has been acquired that Mn-doped CeO_2_performs higher activity than pure CeO_2_ with the same morphology over catalytic oxidation of benzene. Oxygen vacancy is considered to be a critical factor in catalytic performances. For Mn-doped CeO_2_, the existence of Ce-Mn solid solution results in the formation of more oxygen vacancies compared with CeO_2_due to incorporate Mn into CeO_2_ crystal lattice, which can provide more surface active sites to adsorb active oxygen species. This phenomenon has also been identified through the analysis of TPR, in which Mn-doped CeO_2_ possesses more highly reducible surface species closely related with surface oxygen vacancy. In addition, exposed crystal plane can also influence the catalytic activity, which is also attributed to the oxygen vacancy due to the different formation energy of oxygen vacancy on the exposed crystal planes. Based on this, Ce-Mn-NF exhibits higher catalytic ability than Ce-Mn-NC.

As we known, the oxidation of organic molecules over transition metal oxide or mixed metal oxide catalysts involves two identical mechanisms: a Langmuir-Hinshelwood mechanism and a Mars-van Krevelen mechanism [52,53]. At lower temperature, the adsorbed oxygen species with higher activity can enhance the adsorption and oxidation of VOCs, which match with Langmuir-Hinshelwood mechanism. It mainly contains four steps: (1) benzene molecule is adsorbed on the exposed crystal plane to form π-complex of benzene due to the interaction of benzene ring with catalyst; (2) gas-phase oxygen molecule is activated on the crystal plane to adsorb in surface vacancies; (3) the attack of active surface oxygen species to benzene ring; (4) benzene ring breakage and the formation of final products. When the reaction temperature rises, the adsorbed organic molecules are oxidized by the oxygen of metal oxides, which is consistent with the Mars-van Krevelen mechanism. It involves a similar process: (1) adsorption of benzene molecule; (2) the attack of lattice oxygen released from the catalyst to benzene ring; (3) benzene ring breakage and deep oxidation to final products. In addition, there are also two key elements involved in this process: (ⅰ) oxygen vacancies formed are replenished by the surface active oxygen species; (ⅱ) active oxygen species are transported and delivered through the bulk oxygen vacancy (Figure 10). In summary, oxygen vacancy as transport medium, is the link of the whole reaction, therefore it has an important role in determining the catalytic activity.

## 4. Conclusions

Pure CeO_2_ and Mn-doped CeO_2_ with the same morphology (nanofiber and nanocube) were synthesized through hydrothermal method and the complete catalytic oxidation of benzene was examined. The results showed that Mn-doped CeO_2_ possessed higher activity than pure CeO_2_ with the same morphology due to the formation of more oxygen vacancies. In the Mn-doped CeO_2_, Ce-Mn-NF exhibited better catalytic behavior than Ce-Mn-NC, which was closely related the exposed crystal planes. Ce-Mn-NF exposed (110) and (100) planes, while Ce-Mn-NC exposed only (100) plane. According to theoretical calculation, the formation energy of oxygen vacancy on the (110) plane is much less than that on (100), which indicate that oxygen vacancy is easier formed and more surface active species is adsorbed on the (110) plane resulting in the higher catalytic property of Ce-Mn-NF. In general, the effect of crystal plane on the activity is also attributed to the oxygen vacancy. Therefore, oxygen vacancy is a key factor in the process of catalytic reaction.

## Figures and Tables

**Figure 1 molecules-26-06363-f001:**
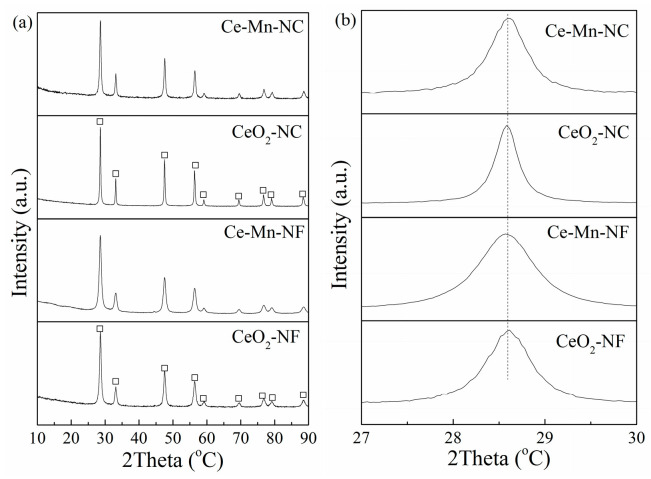
XRD patterns of CeO_2_-NF, CeO_2_-NC, Ce-Mn-NF and Ce-Mn-NC: (**a**) wild angle patterns, and (**b**) Enlarged-zone patterns.

**Figure 2 molecules-26-06363-f002:**
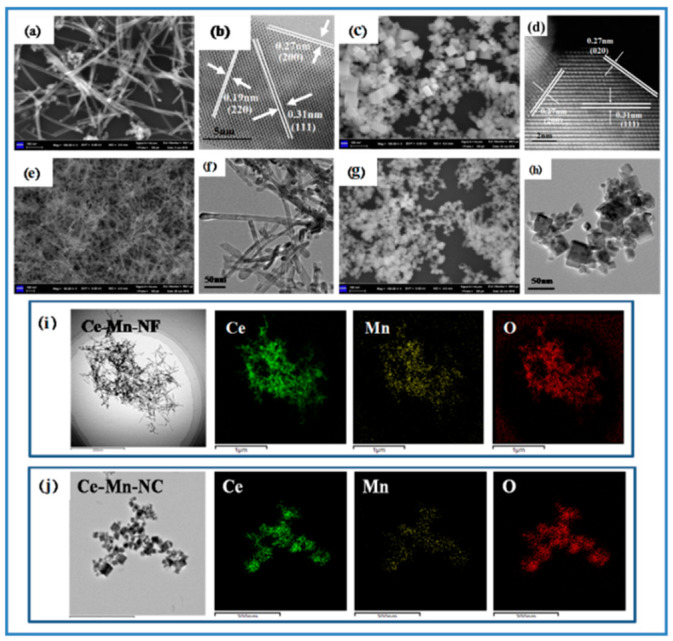
SEM and HRTEM images of CeO_2_-NF and CeO_2_-NC (**a**–**d**); SEM and TEM images of Ce-Mn-NF and Ce-Mn-NC (**e**–**h**); HAADF mapping images of Ce-Mn-NF and Ce-Mn-NC (**i**,**j**).

**Figure 3 molecules-26-06363-f003:**
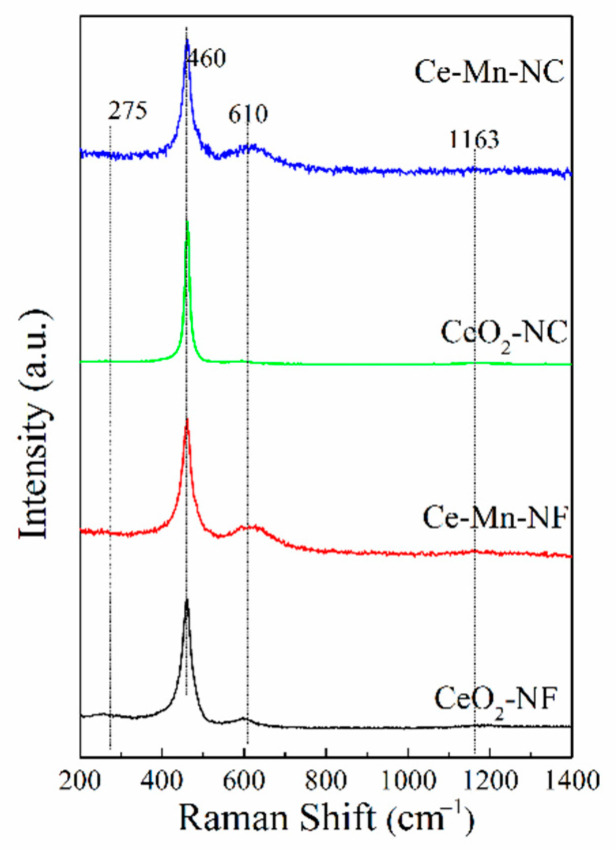
Room temperature visible Raman spectra of all the samples.

**Figure 4 molecules-26-06363-f004:**
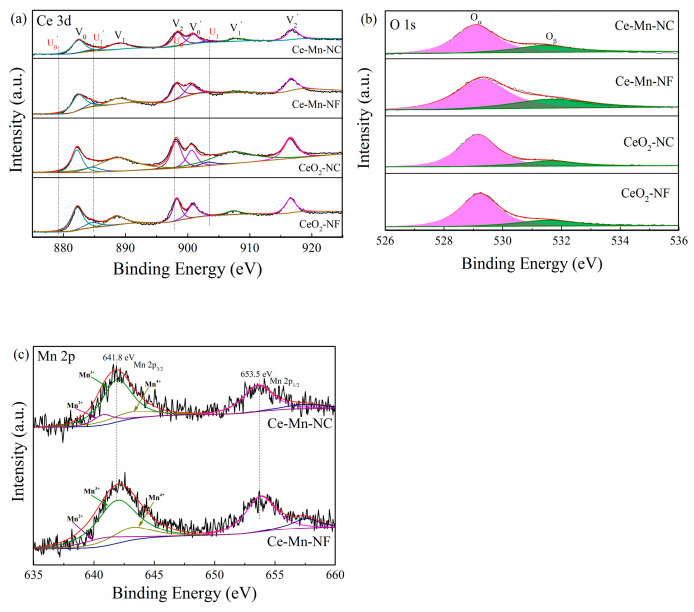
XPS spectra of various catalysts over the spectral regions of Ce 3d (**a**), O 1s (**b**) and Mn 2p (**c**).

**Figure 5 molecules-26-06363-f005:**
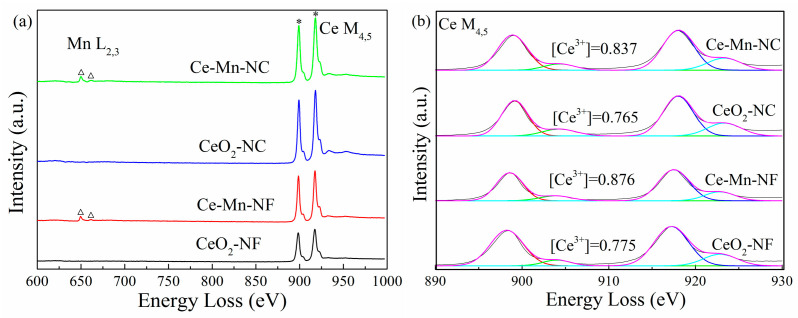
Ce M_4,5_-edge, Mn L_2,3_-edge EELS spectra (**a**) and Ce^3+^ distribution of CeO_2_ samples (**b**).

**Figure 6 molecules-26-06363-f006:**
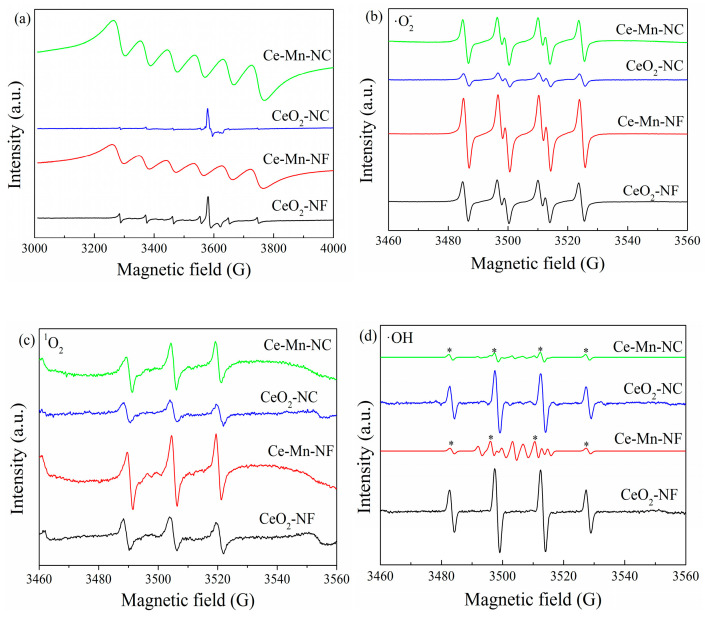
Electron paramagnetic resonance signals of all the samples (**a**), superoxide radical (**b**), singlet oxygen (**c**) and hydroxyl radical (**d**).

**Figure 7 molecules-26-06363-f007:**
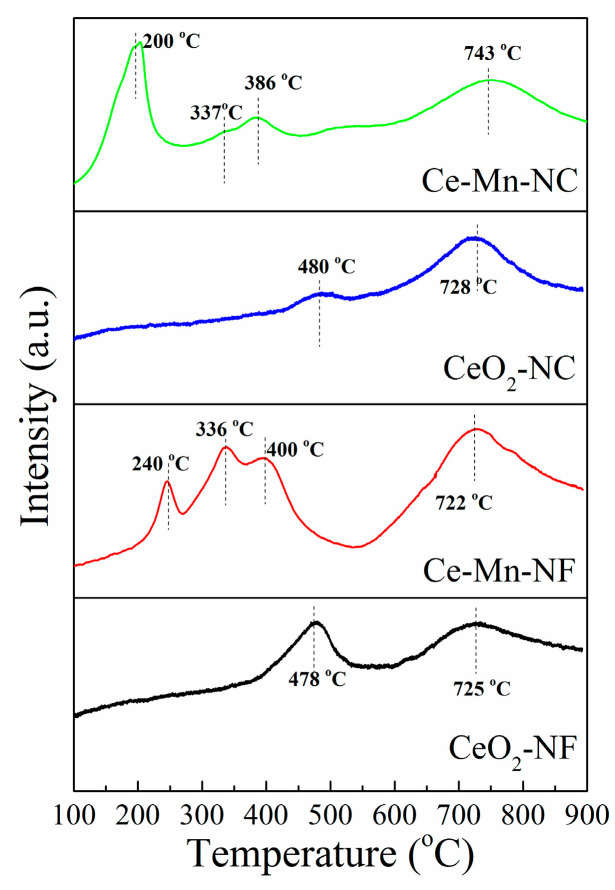
H_2_-TPR profiles of CeO_2_-NF, CeO_2_-NC, Ce-Mn-NF and Ce-Mn-NC.

**Figure 8 molecules-26-06363-f008:**
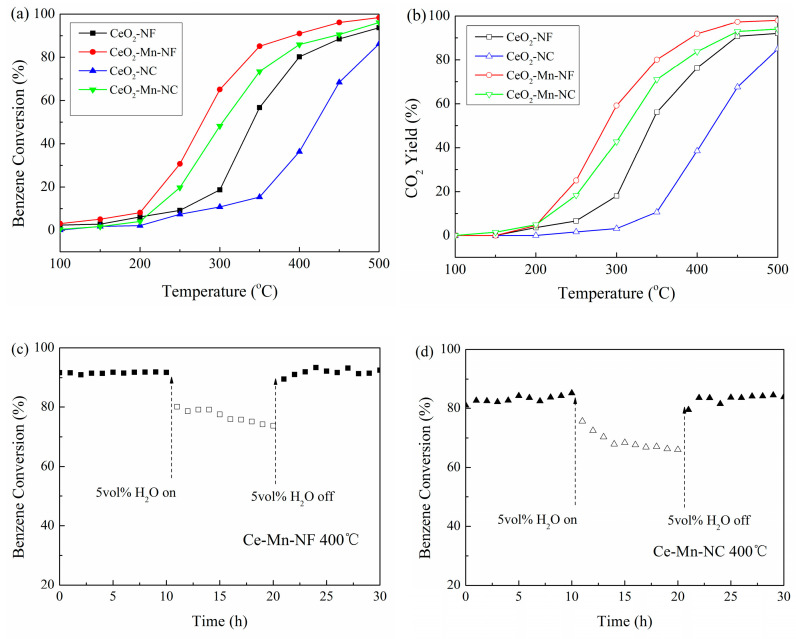
Benzene conversion (%, **a**), CO_2_ yield (%, **b**) over CeO_2_-NF, CeO_2_-NC, Ce-Mn-NF and Ce-Mn-NC. Effect of water vapor on the catalytic activities of Ce-Mn-NF(**c**) and Ce-Mn-NC (**d**). Water concentration = 5 vol%.

**Figure 9 molecules-26-06363-f009:**
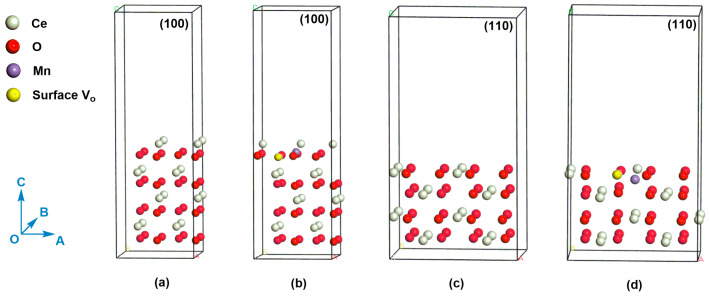
(**a**) (100) slab model for CeO_2_, (**b**) (100) slab model for Mn-doped CeO_2_ with an oxygen vacancy, (**c**) (110) slab model for CeO_2_, (**d**) (110) slab model for Mn-doped CeO_2_ with an oxygen vacancy. The white, red, purple, and yellow atoms refer to Ce atom, O atom, Mn atom, and surface oxygen vacancy, respectively.

**Figure 10 molecules-26-06363-f010:**
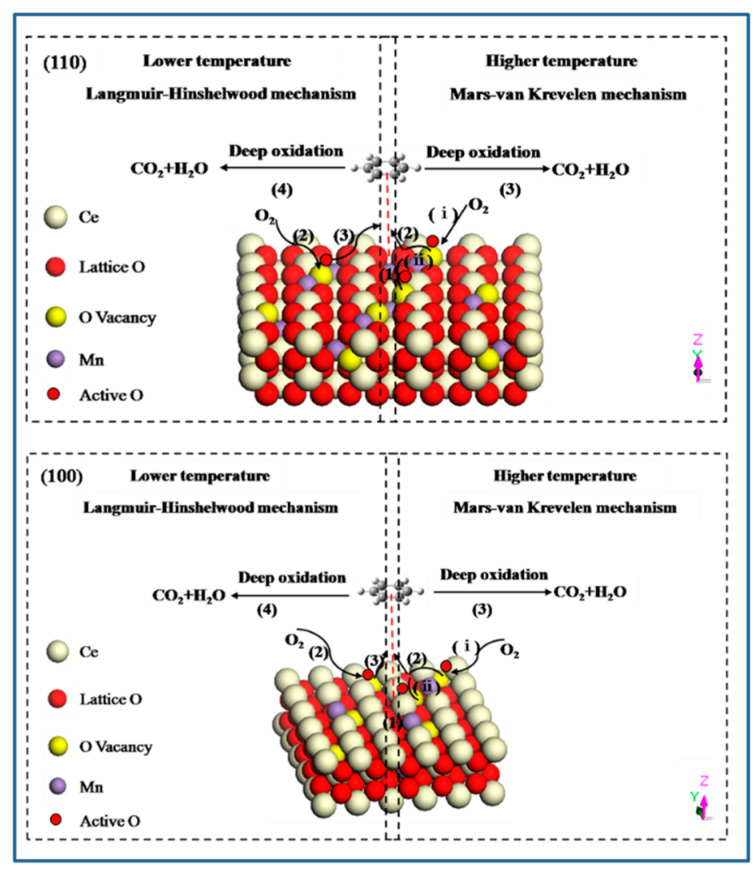
Proposed mechanism for benzene oxidation over Mn-doped CeO_2_ on the different crystal plane.

**Table 1 molecules-26-06363-t001:** Lattice parameters and extent of expansion of synthesized samples compared with those of pure CeO_2_.

Samples	Lattice Parameters (a)	Extent of Deviation
	(Å)	(×10^−4^)
CeO_2_-NF	5.40864	10.06
Ce-Mn-NF	5.40842	11.94
CeO_2_-NC	5.41061	4.02
Ce-Mn-NC	5.40878	11.52

**Table 2 molecules-26-06363-t002:** Physical and chemical properties of all samples.

Sample	A_610_/A_460_(%)	BE (eV)		O_β_/(O_α_ + O_β_)(%)	Ce^3+^/(Ce^3+^ + Ce^4+^) (%)	Mn 2p_3/2_ BE (eV)		
		O_α_	O_β_			Mn^4+^	Mn^3+^	Mn^2+^
CeO_2_-NF	13.9	529.2	531.5	21.1	8.74	----	----	----
Ce-Mn-NF	20.9	529.3	531.7	25.5	11.3	640.9	641.8	643.1
CeO_2_-NC	11.3	529.2	531.6	17.1	7.32	----	----	----
Ce-Mn-NC	19.4	529.1	531.5	22.0	9.71	640.8	641.8	643.2

## Data Availability

Not applicable.

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
