# Peer review of "Roles of Oxygen Vacancies of CeO_2_ and Mn-Doped CeO_2_ with the Same Morphology in Benzene Catalytic Oxidation"

_molecules, 2021, doi:10.3390/molecules26216363_

Round 1
Reviewer 1 Report
I started to review this paper, however I cannot finish it, because of the very bad English. I cannot judge on the manuscript merits, if I do not understand the simple sentences, because of incorrectly used word.
Author Response
We adopt the reviewer’s suggestion. We have invited some experienced colleagues to help with the revision of English throughout the manuscript to make that meets the journal’s desired standard.
Reviewer 2 Report
Manuscript ID: molecules-1403746
Title: Roles of Oxygen Vacancies of CeO2 and Mn-doped CeO2 with the same morphology in Benzene Catalytic Oxidation
Authors: Min Yang, Genli Shen, Qi Wang, Ke Deng, Mi Liu, Yunfa Chen, Yan Gong, Zhen Wang
Dear authors,
This is an interesting work on the decomposition of VOCs by heterogeneous catalysis. The experiments and analysis are fine. I have comments and questions to improve the quality of manuscript.
- The reaction mechanism: The oxidation of hydrocarbons with oxygen involves multiple unit steps. In most cases in autoxidation, the first step of C-H bond dissociation is rate determining. Therefore, how the CeO2 activates the reactant is of interest. This work evaluates the reaction by detecting CO and CO2. Did the authors detect H2O?
- Surface analysis is fine. It would be sounder if the authors can provide the distribution of the vacant site on the surface (similar to chemical mapping).
- The authors consider superoxide anion, singlet oxygen, and hydroxy radical as reactive intermediate based on ESR. Note that singlet oxygens are not free radicals but have different states of electron spin. The description “•1O2” is not common. Whish did the authors detect, “1Δg” or “1∑g+”?
- I understand, from this manuscript, that the key role of the catalyst is to convert triplet-oxygen to singlet-oxygen(s) and hydroxide radical. I could not find a clear explanation from DFT calculation. What is the source of hydroxy radical – contaminated/produced H2O or H in benzene + O in the gas?
Author Response
- The reaction mechanism: The oxidation of hydrocarbons with oxygen involves multiple unit steps. In most cases in auto-oxidation, the first step of C-H bond dissociation is rate determining. Therefore, how the CeO2 activates the reactant is of interest. This work evaluates the reaction by detecting CO and CO2. Did the authors detect H2O?
Answer: Thank the reviewer for the comments. The reaction is evaluated by detecting CO and CO2 in order to investigate the conservation of carbon element. H2O molecule signal and its related information are being tested (such as TPO data, etc). The corresponding results will be shown in subsequent work.
- Surface analysis is fine. It would be sounder if the authors can provide the distribution of the vacant site on the surface (similar to chemical mapping).
Answer: Thank the reviewer for the comments. The suggestion is important to us. The properties of oxygen vacancy on the surface can be reflected indirectly through O1S XPS spectrum. Through analysis, it can be obtained that the relative amount of surface adsorbed oxygen species (Oβ) for Mn doped CeO2 increase compared with that of CeO2, which indicate more surface Ce ions are replaced by Mn to form surface oxygen vacancy. The detailed distribution of oxygen vacancy on the surface will be researched in the following work.
- The authors consider superoxide anion, singlet oxygen, and hydroxy radical as reactive intermediate based on ESR. Note that singlet oxygens are not free radicals but have different states of electron spin. The description “•1O2” is not common. Whish did the authors detect, “1Δg” or “1∑g+”?
Answer: Thank the reviewer for the comments. The suggestion is important to us. “•1O2” in the manuscript have been rewritten as “1O2” according to the literature.
Reference: Environ. Sci. Pollut. Control Ser. 2020, 27, p1230.
Environ. Sci. Technol. 2018, 52, p13405.
- I understand, from this manuscript, that the key role of the catalyst is to convert triplet-oxygen to singlet-oxygen(s) and hydroxide radical. I could not find a clear explanation from DFT calculation. What is the source of hydroxy radical – contaminated/produced H2O or H in benzene + O in the gas?
Answer: Thank the reviewer for the comments. DFT calculation was adopted to calculate the formation energy of oxygen vacancy in order to understand the effect of dopant ions on vacancy formation. It can be acquired that the activated Mn−O and Ce-O can convert surface adsorbed O2/H2O into reactive oxygen species like •OH and •O2− according to relative literature.
Reference: Environ. Sci. Technol. 2018, 52, p13405.
Round 2
Reviewer 1 Report
Please submit corrected manuscript without track changes.

Author Response
I have submitted the corrected manuscript without track changes
Reviewer 2 Report
Manuscript ID: molecules-1403746
Title: Roles of Oxygen Vacancies of CeO2 and Mn-doped CeO2 with the same morphology in Benzene Catalytic Oxidation
Authors: Min Yang, Genli Shen, Qi Wang, Ke Deng, Mi Liu, Yunfa Chen, Yan Gong, Zhen Wang
Dear authors,
Thank you for the update. In my opinion, this is acceptable.
Author Response
There are no further changes in the revised manuscript according to reviewer's comment during the second round of review